DATA RELEASE

# Insecticide resistance dynamics in *Anopheles gambiae* s.l. in Ghana

Christopher Mfum Owusu-Asenso[1], Anisa Abdulai[1], Isaac Kwame Sraku[1], Yasmeen Amandi Acquah[1], Stephina Adjoa Yanney[1], Miriam DedeAma Dortey[1], Yaw Akuamoah-Boateng[1], Abdul Rahim Mohammed Sabtiu[1], Judith Dzifa Azumah[1], Abena Ahema Ebuako[1], Nutifafa Efui Abusa[1], Nana Kwame Baako[1], Christabel Asabea Koranteng[1], Ruth Owusu Kwarteng[1], Grace Arhin Danquah[1], Godfred Amoateng[1], Bright Churchill Obeng[1], Akua Obeng Forson[2], Cornelia Appiah-Kwarteng[3], Simon Kwaku Attah[1] and Yaw Asare Afrane[1,*]

1 Centre for Vector-Borne Disease Research, Department of Medical Microbiology, University of Ghana Medical School, Korle-Bu, Accra, Ghana
2 Department of Medical Laboratory Science, School of Biomedical and Allied Health Sciences, University of Ghana, Korle-Bu, Accra, Ghana
3 School of Veterinary Medicine, University of Ghana, Legon, Accra, Ghana

## ABSTRACT

Malaria control in Ghana and sub-Saharan Africa is threatened by widespread insecticide resistance in *Anopheles gambiae* s.l., undermining the effectiveness of long-lasting insecticidal nets and indoor residual spraying. A longitudinal survey was conducted between 2023 and 2025 across 20 urban and suburban sites spanning the coastal savannah, forest, and Sahel savannah zones. Of the 1,008 *An. gambiae* s.l. sampled, *An. coluzzii* was the dominant species (65.1%), followed by *An. gambiae* s.s. (18.9%) and *An. arabiensis* (10.9%). WHO bioassays revealed high pyrethroid resistance (mortality rate = 20–45%). Full susceptibility to pirimiphos-methyl (mortality rate = 99–100%) and chlorfenapyr was observed at most sites, though resistance to clothianidin was observed in Obuasi, Tema, and Abossey Okai. Intensity assays confirmed strong pyrethroid resistance even at 10× diagnostic concentrations. Genotyping showed near-fixation of the *kdrL995F* allele and the presence of additional resistance markers, including *N1570Y, V402L, I1527T,* and Ace-1R *G280S.*

**Submitted:** 22 September 2025

* Corresponding author. Email: yafrane@ug.edu.gh

Preprint submitted at https://doi.org/10.60763/africarxiv/10162

Included in the series: *Vectors of human disease* (https://doi.org/10.46471/GIGABYTE_SERIES_0002)

**Subjects** Ecology, Biodiversity, Taxonomy

# DATA DESCRIPTION
## Background and context

Insecticide resistance in malaria vectors poses a critical challenge to global efforts to control and eliminate malaria [1, 2]. While the widespread deployment of long-lasting insecticidal nets and indoor residual spraying has led to a substantial reduction in malaria transmission, the emergence of resistance across all six major classes of public health insecticides threatens to reverse these gains [3]. In Ghana, where malaria remains endemic, widespread pyrethroid resistance and the emergence of resistance to newer insecticide classes, such as neonicotinoids and pyrroles, have been increasingly reported [3]. Despite continuous national efforts to combat resistance, such as rotating insecticide classes and

deploying synergist-based interventions (e.g., piperonyl butoxide nets), the underlying drivers and mechanisms of resistance remain poorly understood.

The evolution of insecticide resistance in *Anopheles gambiae* s.l. may be driven by both selection pressures from public health insecticides for vector control and environmental exposures, including the extensive use of insecticides in agriculture [4–7]. These resistance mechanisms vary spatially and temporally, and could be influenced by ecological factors, anthropogenic activities, and historical patterns of insecticide deployment.

Despite growing evidence of widespread resistance, comprehensive data on the phenotypic and genotypic resistance of *An. gambiae* s.l. populations remain limited. There is thus an urgent need to characterize the full resistance profile of vector populations across different ecological zones to inform evidence-based control strategies. This study provides data on the dynamics of *An. gambiae* s.l. resistance across diverse ecological settings in Ghana between 2023 and 2025. By integrating insecticide susceptibility testing and genotyping of resistance alleles, this study provides a comprehensive assessment of resistance mechanisms driving reduced insecticide efficacy in Ghanaian vector populations.

## METHODS

This longitudinal study was conducted between January 2023 and December 2025 across 16 selected urban (Takoradi, Tema, Accra, Teshie Tuba, Tamale) and suburban (Ada, Dodowa, Elubo, Kpalsogu, Taha, Kulaa, Tolon, Libga Obuasi, Konongo) sites in Ghana (Figure 1). Study sites were chosen to represent diverse ecological zones.

The coastal savannah in southern Ghana has a tropical savannah climate with temperatures ranging from 23 to 34 °C and an average annual rainfall of 787 mm, following a bimodal pattern with rainy seasons from April to July and from September to November. The dry season lasts from December to March. The forest zone in the middle of Ghana experiences a tropical forest climate with 1,500 to 2,000 mm of annual rainfall, also in a bimodal pattern, with rainy seasons from March to July and September to November, and dry periods from December to February. Temperatures remain stable between 24 and 30 °C. The Sahel savannah in the north of Ghana has a unimodal rainfall pattern from May to November, averaging 900 mm annually. The dry season (December–April) sees temperatures rising to 42 °C, with a mean annual temperature of 28 °C.

### Sampling of *Anopheles gambiae* s.l. and rearing in the insectary

*Anopheles* larvae sampling was carried out from January 2023 to July 2025. To avoid collecting members from a single population, larvae were sampled randomly from different breeding habitats at each study site and then pooled. Immature *Anopheles* mosquitoes collected from the same site within the Sahel-savannah zone were transported to the insectary of the President's Malaria Initiative Project Office in Tamale. Similarly, larvae collected from sites within the forest and coastal savannah ecozones were transported to the insectary of the AngloGold Ashanti malaria control programme in Obuasi, and to the Department of Medical Microbiology at the University of Ghana Medical School, Korle-Bu, Accra, respectively. The mosquito larvae were raised to adults at the insectary, maintained at an average temperature of 28 ± 1 °C and a relative humidity of 80.9 ± 6.3%. Once emerged, the adult mosquitoes were provided with a 10% sucrose solution for feeding. Three to five days old, non-blood-fed females were selected for downstream World Health Organization (WHO) phenotypic bioassays and genotypic assays. All mosquitoes used for susceptibility



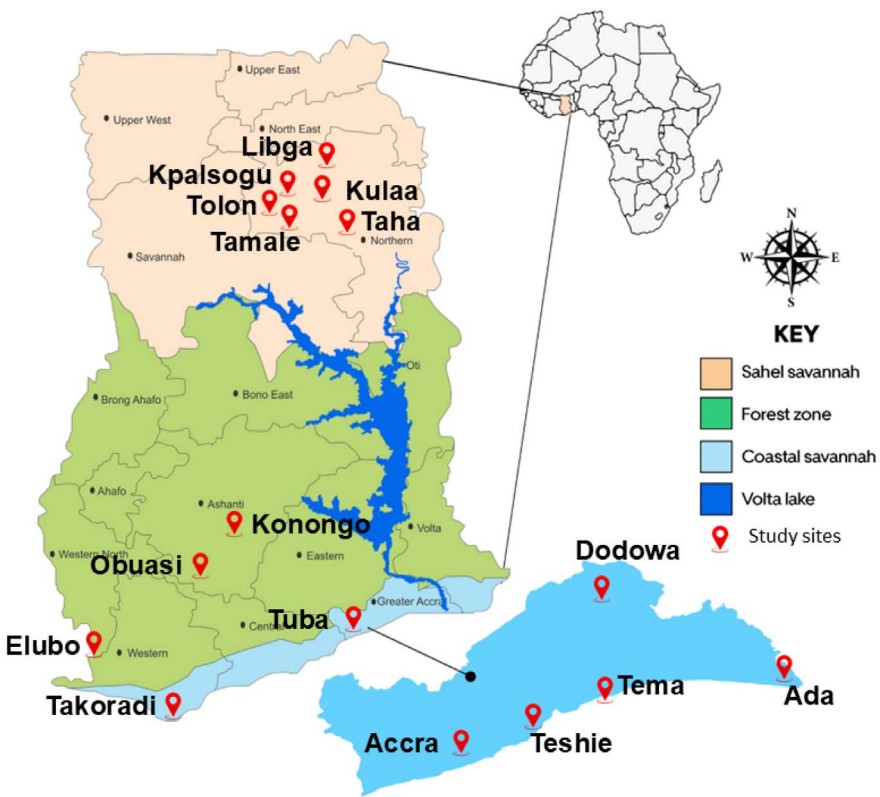

**Figure 1.** Map of Ghana showing the various study sites.

testing were morphologically identified to confirm *Anopheles gambiae* s.l., ensuring that only the target vector population was included in the assays.

## Assessment of phenotypic resistance in *Anopheles gambiae* s.l. using the WHO standard and intensity bioassay

To assess the resistance intensity of *An. gambiae* s.l. population across various site categories, a batch of 25 non-blood-fed females aged 3–5 days were subjected to the WHO susceptibility test. Four replicates and two controls were used for each insecticide using the standard WHO tube assay procedure [8, 9]. The WHO test papers impregnated with insecticides at discriminating concentrations used were: permethrin [1× (0.75%), 5× (3.75%), and 10× (7.5%)], deltamethrin [1× (0.05%), 5× (0.25%), and 10× (0.5%)], and pirimiphos-methyl [1× (0.25%), 5× (1.25%), and 10× (2.5%)] [9]. Mosquitoes were not exposed to 2× insecticide concentration due to logistical constraints and the focus on assessing high-intensity resistance, as recommended by the WHO for populations showing high resistance. The Knocked-down mosquitoes were recorded at the end of the 60 min exposure period. After exposure, mosquitoes were transferred to a paper cup covered with untreated netting and provided with a 10% sugar solution soaked in a wad of cotton. Mortalities were recorded after 24 hours. The insecticide papers were initially tested on the Kisumu susceptible strain, a reference strain susceptible to insecticides, to confirm their efficacy.

### Phenotypic resistance to chlorfenapyr and clothianidin in *Anopheles gambiae* s.l. using the WHO bottle bioassay

To determine the susceptibility status of *Anopheles gambiae* s.l. to chlorfenapyr and clothianidin, WHO Bottle bioassays were carried out using four replicates according to WHO guidelines [10]; each replicate consisted of 25 female *Anopheles* mosquitoes aged 3–5 days. These mosquitoes were exposed to 100 µg/ml chlorfenapyr-acetone solution and 4 µg/ml clothianidin-acetone-MERO solution for 1 h each. This was done approximately 24 h after coating the bottles with the respective insecticides, to allow proper drying. Two additional replicates of 25 mosquitoes each served as negative controls, with bottles treated with 1 ml of acetone for chlorfenapyr or with acetone containing MERO for clothianidin. The knockdown of mosquitoes was recorded at the end of the 60-minute exposure period. After exposure, mosquitoes were transferred to paper cups covered with untreated netting and provided with a 10% sugar solution soaked in a cotton wad, which was replaced daily. Mortality was monitored and recorded at 24, 48, and 72 hours for chlorfenapyr, while for clothianidin mortality was recorded after 24 hours.

### Detection of target-site mutations in *An. gambiae* s.l.

Conventional and real-time PCR were used to investigate the presence of insecticide resistance genes, including *kdr haplotypes* (*L995F, L995S, N1570Y, V402L*, and *I1527T*) and *Ace-1$^R$ G280S* [11]. The allele-specific PCR procedure for *kdr* genotyping was used to detect *kdr* alleles using the protocol and primer sequences described by Martines-Torres *et al.* [12] and Jones *et al.* [13] for *N1570Y*; Williams *et al.* [14] for *V402L*, I1527T, and *P1874L*; and Weil *et al.* [15] for the *Ace-1$^R$ G280S* mutation.

### Characterization of *Anopheles gambiae* s.l.

A sub-sample of the *An. gambiae* s.l., after the phenotypic susceptibility testing, were randomly selected and identified morphologically using the keys of Coetzee [16]. Members of the *An. gambiae* s.l. were further identified by PCR to distinguish sibling species using a leg of each mosquito, as previously described by Scott *et al.* [17].

### Data management and analysis

Descriptive analysis was conducted to visualize WHO susceptibility data, resistant allele frequencies, and mosquito species composition from the selected sites using graphs and tables. WHO insecticide susceptibility levels were classified using the WHO criteria [10, 18]. Allele frequencies of resistance gene markers in the vector populations at each site were calculated using the Hardy–Weinberg equilibrium, with the formula F (allele frequency) = (2nRR + nRS)/2$N$, where nRR is the number of individuals homozygous for the resistant allele (RR), nRS is the number of heterozygous individuals (RS), and $N$ is the total number of individuals successfully genotyped in the population. In all analyses, a $P$-value ≤ 0.05 was considered statistically significant.

### DATA VALIDATION AND QUALITY CONTROL

Mosquitoes were identified by experienced taxonomists using standard morphological keys [16, 19] and confirmed by molecular assays [17]. Insecticide susceptibility bioassays were conducted following WHO protocols [10, 20], with replicates and controls included to ensure accuracy. Genotyping assays incorporated positive and negative controls



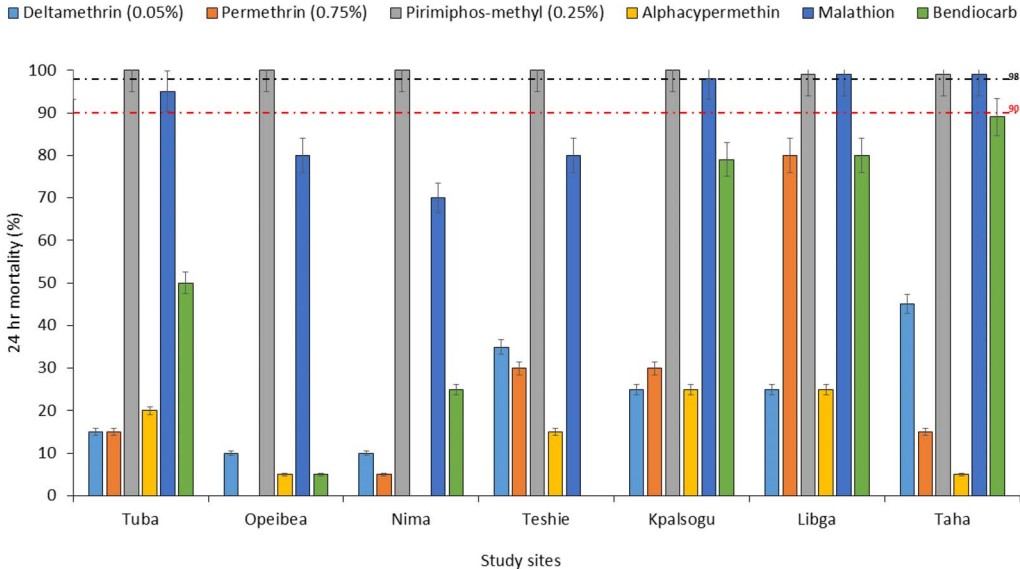

**Figure 2.** WHO insecticide susceptibility bioassay results across different study sites.

[12, 15, 21, 22], and a subset of samples was re-tested to confirm consistency. All data were cleaned, double-checked for errors, and cross-validated prior to statistical analysis.

The final dataset was published and validated through the Integrated Publishing Toolkit of the Global Biodiversity Information Facility (GBIF) [23], which ensures structural and content quality checks before publication. Metadata fields are available on the dataset page in GBIF, providing transparency and facilitating reuse [23].

## RESULTS

### Phenotypic resistance in *Anopheles gambiae* s.l. using the WHO standard, intensity and bottle bioassays

Resistance to pyrethroids was widespread, across all study sites: deltamethrin mortality rate (*MR*) = (10–45%), permethrin (*MR* = 0–80%), and alphacypermethrin (*MR* = 0–25%). Vectors were susceptible to pirimiphos-methyl, with *MR* = 99–100%. Resistance to susceptibility was observed in vectors exposed to malathion (*MR* = 70–99%). Resistance to Bendiocarb (0–89%) was observed in vectors across all sites (Figure 2).

Intensity assays revealed high-pyrethroid resistance across all sites: deltamethrin *MR* = 66–96% (10×), permethrin *MR* = 74–95% (10×). Moderate resistance to full susceptibility was observed in vectors exposed to pirimiphos-methyl (*MR* = 92–100%); however, high resistance was observed in vectors from Obuasi (*MR* = 0–2%) (Figure 3).

Across all study sites, *Anopheles gambiae* s.l. populations exhibited generally high mortality to both chlorfenapyr and clothianidin, although site-specific variations were observed (Figure 4). *Anopheles* vectors showed full susceptibility to chlorfenapyr at Kokompe, Tamale Fitam, Tema, and Accra Industrial Area (*MR* = 100%). However, resistance and possible resistance were observed in vectors from Abossey Okai (*MR* = 80–84%) and Obuasi (*MR* = 95–97%), respectively (Figure 4).

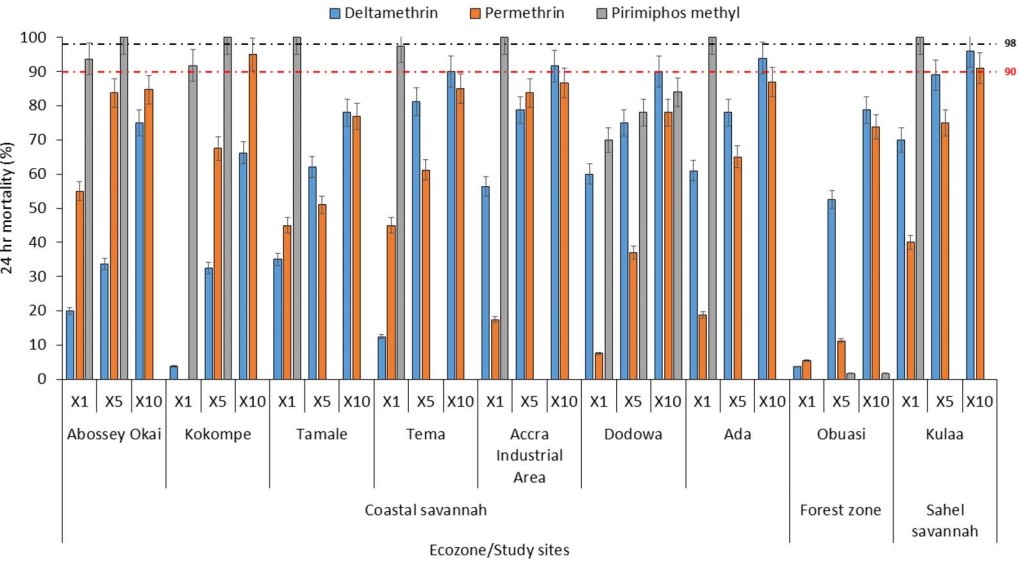

**Figure 3.** WHO intensity bioassay at different study sites across varied ecozones.

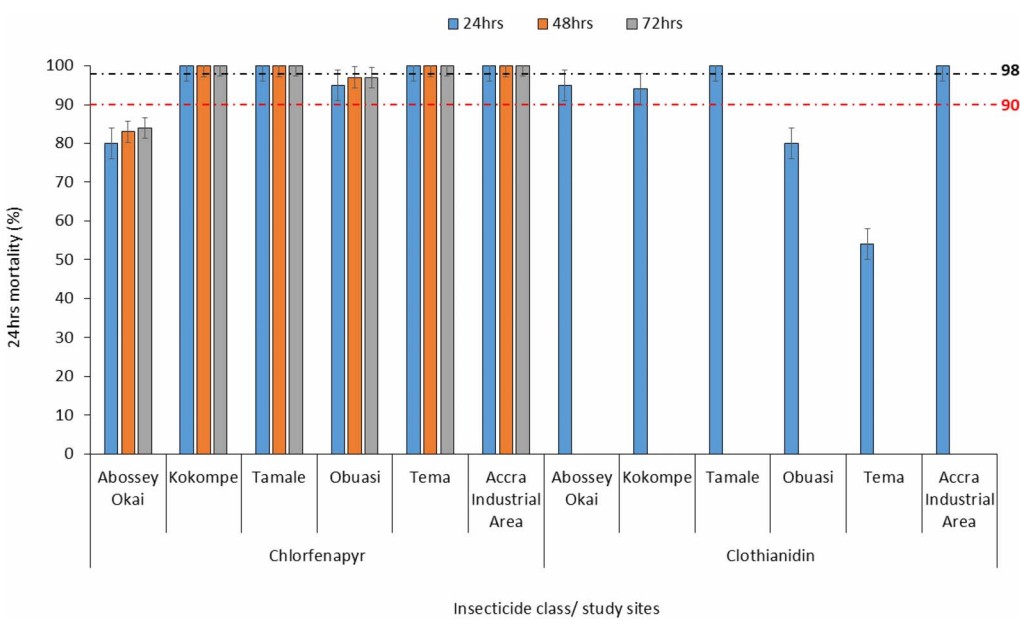

**Figure 4.** Results of the WHO bottle bioassay of insecticide susceptibility for chlorfenapyr and clothianidin in *Anopheles gambiae* s.l.

Susceptibility to clothianidin was observed in vectors from Tamale Fitam, and Accra Industrial Area (*MR* = 100%), whereas vectors from Obuasi (80%) and Tema Community 1 (*MR* = 54%) were resistant. *Anopheles gambiae* s.l. populations from Abossey Okai (*MR* = 95%) and Kokompe (*MR* = 94%) showed suspected resistance when exposed to clothianidin (Figure 4).



**Table 1.** Species distribution of *Anopheles* mosquitoes across study sites in Ghana.

| Study site | *An. arabiensis* | *An. coluzzii* | *An. gambiae* s.s. | *An. melas* | Hybrid | *An. gambiae* (unidentified) | Total |
|---|---|---|---|---|---|---|---|
| Accra | 0 | 201 | 111 | 0 | 5 | 17 | 334 |
| Ada | 0 | 28 | 1 | 1 | 0 | 0 | 30 |
| Dodowa | 0 | 21 | 1 | 0 | 0 | 8 | 30 |
| Elubo | 0 | 35 | 1 | 0 | 0 | 4 | 40 |
| Konongo | 0 | 40 | 0 | 0 | 0 | 0 | 40 |
| Kpalsogu | 0 | 27 | 10 | 0 | 0 | 6 | 43 |
| Kulaa | 54 | 6 | 8 | 0 | 1 | 0 | 69 |
| Libga | 6 | 32 | 4 | 0 | 0 | 2 | 44 |
| Obuasi | 0 | 28 | 1 | 0 | 1 | 0 | 30 |
| Taha | 37 | 4 | 1 | 0 | 0 | 1 | 43 |
| Takoradi | 0 | 30 | 10 | 0 | 0 | 0 | 40 |
| Tamale | 10 | 8 | 9 | 0 | 3 | 0 | 30 |
| Tema | 0 | 110 | 0 | 0 | 0 | 0 | 110 |
| Teshie | 0 | 38 | 9 | 0 | 0 | 0 | 47 |
| Tolon | 3 | 18 | 19 | 0 | 0 | 0 | 40 |
| Tuba | 0 | 30 | 6 | 0 | 0 | 2 | 38 |
| **Total** | **110** | **656** | **191** | **1** | **10** | **40** | **1,008** |

**Table 2.** Distribution of *Anopheles gambiae* species across years.

| Species | 2023 | 2024 | 2025 | Total |
|---|---|---|---|---|
| *An. arabiensis* | 107 | 3 | 0 | 110 |
| *An. coluzzii* | 297 | 205 | 174 | 676 |
| *An. gambiae* s.s. | 96 | 36 | 67 | 199 |
| *An. melas* | 0 | 1 | 22 | 23 |
| *Hybrid* | 7 | 5 | 2 | 14 |
| **Total** | **520** | **250** | **238** | **1,008** |

## Species discrimination in *Anopheles gambiae* s.l.

Genotyping of 1,008 *An. gambiae* s.l. showed that the most abundant species were
*An. coluzzii* ($n$ = 673; 66.77%), followed by *An. gambiae* s.s. ($n$ = 210; 20.83%), *An. arabiensis*
($n$ = 110; 10.91%), *An. melas* ($n$ = 1; 0.10%), *hybrids* ($n$ = 14; 1.39%), and *Anopheles gambiae* s.l.
*unidentified* ($n$ = 14; 1.4%) (Table 1).

## Temporal distribution of *An. gambiae* s.l.

*Anopheles coluzzii* remained the predominant species throughout the sampling years,
accounting for 57.1% ($n$ = 297) of *Anopheles* mosquitoes collected in 2023, 82.0% ($n$ = 205) in
2024, and 64.7% ($n$ = 154) in 2025. This was followed by *An. gambiae* s.s. (from 18.5% in 2023
to 24.8% in 2025). *Anopheles arabiensis* was almost exclusively collected in 2023 ($n$ = 107),
with only three species sampled in 2024 and none in 2025, suggesting strong temporal
heterogeneity in its occurrence. *Anopheles melas* was observed only at coastal sites sampled
in 2024 and 2025, while hybrid forms were rare, comprising <2% of the total collections
across the study period (Table 2).

Overall, *An. coluzzii* (64.48%) and *An. gambiae* s.s. (18.65%) were the most abundant, with
both species strongly associated with urban settings. In contrast, *An. arabiensis* was more
evenly distributed, occurring predominantly in suburban (41.8%) sites. *Anopheles melas*
was restricted to urban areas (1.39%), while *hybrid* forms (1.39%) were rare but detected in
both suburban and urban environments (Figure 5).

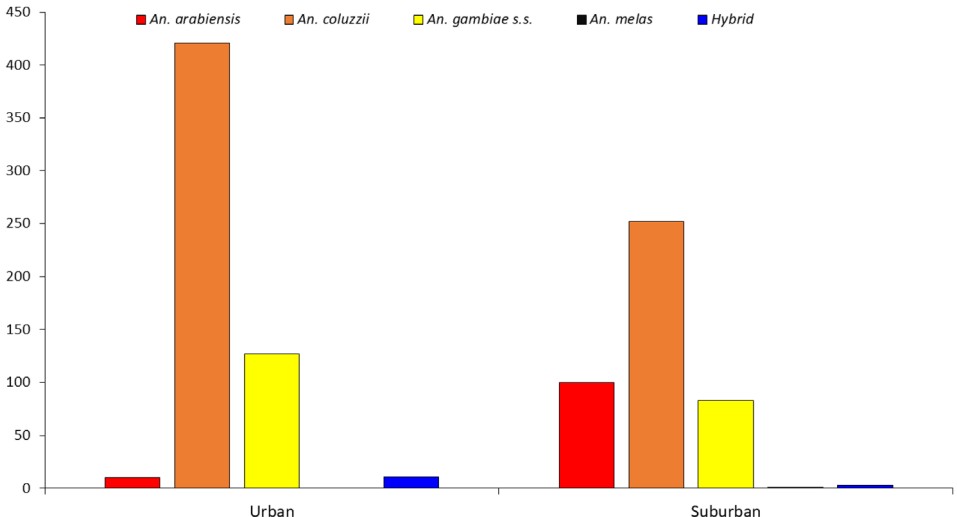

**Figure 5.** Species composition of the *Anopheles gambiae* complex in urban and suburban study populations.

**Table 3.** Resistance mutation genotyping across study sites.

| Study site | L995F | L995S | N1570Y | G280S | V402L | I1527T |
|---|---|---|---|---|---|---|
| Elubo | 0.512 | 0.025 | 0.141 | - | 0.163 | 0.125 |
| Konongo | 1.000 | - | - | - | 0.438 | 0.013 |
| Kpalsogu | 1.000 | 0.342 | 0.109 | 0.000 | 0.000 | 0.000 |
| Kulaa | 1.000 | 0.207 | 0.104 | - | - | - |
| Libga | 1.000 | 0.159 | 0.109 | 0.000 | 0.000 | 0.000 |
| Accra | 1.000 | 0.250 | 0.433 | 0.333 | 0.467 | 0.500 |
| Tema | 0.117 | 0.675 | - | 0.000 | 0.000 | 0.000 |
| Teshie | 1.000 | 0.022 | 0.043 | 0.000 | 0.000 | 0.000 |
| Tolon | 0.717 | 0.012 | 0.160 | 0.000 | 0.094 | 0.013 |
| Tuba | 1.000 | 0.013 | - | 0.000 | 0.000 | 0.000 |
| Taha | 1.000 | 0.093 | 0.163 | 0.000 | 0.000 | 0.000 |
| Takoradi | 0.713 | 0.025 | 0.269 | 0.000 | 0.400 | 0.100 |
| Ada | 0.983 | 0.123 | 0.383 | 0.391 | 0.679 | 0.417 |
| Obuasi | 0.967 | 0.258 | 0.496 | 0.400 | 0.713 | 0.550 |
| Tamale Fitam | 0.880 | 0.111 | 0.320 | 0.095 | 0.750 | 0.408 |
| Dodowa | 0.933 | 0.067 | 0.490 | 0.370 | 0.960 | 0.300 |

-: not determined.

## Genotyping of target site mutations in *Anopheles gambiae* s.l.

*kdr* genotyping revealed multiple insecticide resistance mutations across study sites (Table 3). The *kdr L1014F* mutation was the most widely distributed, with allele frequencies ranging from 0.51 in Elubo to fixation (1.0) in several sites, including Konongo, Kpalsogu, Kulaa, Libga, Accra, Tuba, Taha, and Takoradi. The *kdr L1014S* allele showed lower and more variable frequencies (0.16–0.34). Detailed results are shown in Table 3.

## RE-USE POTENTIAL

The characterisation of both phenotypic and genotypic insecticide resistance in malaria vectors provides critical support for evidence-based vector control decision-making [5, 24]. In this study, *Anopheles gambiae* s.l. populations sampled across diverse ecological settings

in Ghana exhibited high levels of pyrethroid resistance, as corroborated by bioassay outcomes and the detection of key resistance-associated mutations, including *kdr* variants (*L995F, L995S, N1570Y, V402L*, and *I1527T*) and *Ace-1 G280S*. These findings are similar to other studies from Ghana that reported widespread insecticide resistance [25, 26] and pose a sustained threat to the effectiveness of current vector control tools.

The integration of phenotypic resistance intensity data with underlying molecular mechanisms enhances the interpretability and utility of the dataset. Such data can be readily reused for longitudinal and comparative analyses to track spatiotemporal trends in resistance, evaluate the spread of resistance alleles, and assess the impact of environmental and operational factors on resistance development [5]. In addition, the dataset can support the calibration of predictive models aimed at forecasting intervention performance across varying resistance scenarios.

Beyond local application, these data have broader relevance for regional and continental resistance surveillance efforts. When integrated into national and global databases, the findings can inform resistance management strategies, guide insecticide rotation and deployment policies, and support the refinement of integrated vector management frameworks [27–29]. Overall, this dataset strengthens surveillance of malaria vector resistance and provides a valuable resource for researchers, public health practitioners, and policy-makers working toward sustainable malaria control in Africa.

## DATA AVAILABILITY

The data supporting this article are published through the Integrated Publishing Toolkit of GBIF and are available under a CC0 waiver from GBIF [23].

## EDITOR'S NOTE

This paper is part of a series of Data Release articles working with GBIF and supported by TDR, the Special Program for Research and Training in Tropical Diseases, hosted at the World Health Organization [30].

## ABBREVIATIONS

GBIF, Global Biodiversity Information Facility; MR, mortality rate; WHO, World Health Organization.

## DECLARATIONS

### Ethics approval and consent to participate

This study received scientific and ethical approval from the Ethical and Protocol Review Committee (EPRC) of the College of Health Sciences, University of Ghana, Korle-Bu Campus. Permission to conduct the study at the various sites was obtained from the community leaders prior to conducting mosquito sampling at all study locations.

### Competing interests

The authors declared no conflict of interest.

### Authors' contributions

CMO-A, AA, AOF, CA-K, SKA and YAA were responsible for the study design, supervised the data collection, and contributed to the writing of the manuscript. CMO-A, AA, ARMS, IKS,

YAB, YAA, SAY, MDD, YA-B, JDA, AAE, NEA, NKB, CAK, ROK, GDA, GA and CBO performed the data collection and laboratory analysis. CMO-A performed the data analysis and visualization, and drafted the manuscript. All the authors read and approved the final manuscript.

## Funding

This study was supported by grants from the National Institute of Health received by YAA (Grant numbers: R01 A1123074, R03 AI186018, and D43 TW011513). The funders had no role or influence on the design of this study, data collection, analyses, and interpretation of the data collected, as well as in writing this manuscript.

## Acknowledgements

We are grateful to the community members and field assistants for their support during sample collection and for granting permission to conduct the study within their localities.

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
