## [Editor Report]

Editor’s AssessmentIn Sub Saharan Africa malaria is a major public health challenge accounting for over 251 million cases and over 500,000 deaths, there is however limited data on distribution of insecticide resistance, which hinders modelling research and control strategies. This paper is one of a series of Data Release papers in GigaByte supported by TDR and the WHO describing datasets hosted in GBIF to tackle these data gaps in vectors of human disease data. This paper presents a longitudinal survey was conducted between 2023 and 2025 across 20 urban and suburban sites spanning the coastal savannah, forest, and Sahel savannah zones. WHO bioassays revealed high pyrethroid resistance. Peer review and data auditing found the data to be well validated. The information contained can serve as a resource for studies focused on assessing transmission risks, vector control strategies, disease surveillance and a broader comprehension of Anopheles mosquito ecology and insecticide resistance across Ghana.Editor’s AssessmentIn Sub Saharan Africa malaria is a major public health challenge accounting for over 251 million cases and over 500,000 deaths, there is however limited data on distribution of insecticide resistance, which hinders modelling research and control strategies. This paper is one of a series of Data Release papers in GigaByte supported by TDR and the WHO describing datasets hosted in GBIF to tackle these data gaps in vectors of human disease data. This paper presents a longitudinal survey was conducted between 2023 and 2025 across 20 urban and suburban sites spanning the coastal savannah, forest, and Sahel savannah zones. WHO bioassays revealed high pyrethroid resistance. Peer review and data auditing found the data to be well validated. The information contained can serve as a resource for studies focused on assessing transmission risks, vector control strategies, disease surveillance and a broader comprehension of Anopheles mosquito ecology and insecticide resistance across Ghana.

---

## [Reviewer Report]

Reviewer name and names of any other individual's who aided in reviewer YannanDo you understand and agree to our policy of having open and named reviews, and having your review included with the published papers. (If no, please inform the editor that you cannot review this manuscript.)YesIs the language of sufficient quality?YesPlease add additional comments on language quality to clarify if needed
Are all data available and do they match the descriptions in the paper? YesAdditional CommentsAre the data and metadata consistent with relevant minimum information or reporting standards? See GigaDB checklists for examples <a href="http://gigadb.org/site/guide" target="_blank">http://gigadb.org/site/guide</a>YesAdditional CommentsIs the data acquisition clear, complete and methodologically sound?YesAdditional CommentsIs there sufficient detail in the methods and data-processing steps to allow reproduction?YesAdditional CommentsIs there sufficient data validation and statistical analyses of data quality? YesAdditional CommentsIs the validation suitable for this type of data?YesAdditional CommentsIs there sufficient information for others to reuse this dataset or integrate it with other data?YesAdditional CommentsPlease update GBIF reference 24 to: Afrane Y A, Owusu-Asenso C M, Akuamoah-Boateng Y, Abdulai A, Mohammed Sabtui A R (2025). Phenotypic and Genotypic Insecticide Resistance Profiles of Anopheles Mosquitoes Across Ghana. Version 1.3. University of Ghana Medical School. Occurrence dataset https://doi.org/10.15468/pgpz3d accessed via GBIF.org on 2025-10-11.Any Additional Overall Comments to the AuthorRecommendationAccept

---

## [Reviewer Report]

Upload additional filesDRR-202509-04-R01/stage_files/DRR-202509-04/Review MS/IR_Dynamics_in_An._gambiae_Ghana.pdfReviewer name and names of any other individual's who aided in reviewer Udoka NwangwuDo you understand and agree to our policy of having open and named reviews, and having your review included with the published papers. (If no, please inform the editor that you cannot review this manuscript.)YesIs the language of sufficient quality?YesPlease add additional comments on language quality to clarify if needed
Are all data available and do they match the descriptions in the paper? YesAdditional CommentsAre the data and metadata consistent with relevant minimum information or reporting standards? See GigaDB checklists for examples <a href="http://gigadb.org/site/guide" target="_blank">http://gigadb.org/site/guide</a>YesAdditional CommentsIs the data acquisition clear, complete and methodologically sound?YesAdditional CommentsIs there sufficient detail in the methods and data-processing steps to allow reproduction?YesAdditional CommentsIs there sufficient data validation and statistical analyses of data quality? YesAdditional CommentsIs the validation suitable for this type of data?YesAdditional CommentsIs there sufficient information for others to reuse this dataset or integrate it with other data?YesAdditional CommentsAny Additional Overall Comments to the AuthorThe manuscript is well-written, technically sound, and presents valuable data. However, minor corrections are needed, and the manuscript would benefit from including interpretative discussion, particularly within the “Re-use Potential” section, to better contextualize the findings and highlight their broader implications.RecommendationMinor Revision